# Effects of Kinesio Taping and Rigid Taping on Gluteus Medius Muscle Activation in Healthy Individuals: A Randomized Controlled Study

**DOI:** 10.3390/ijerph192214889

**Published:** 2022-11-12

**Authors:** Kamil Zaworski, Joanna Baj-Korpak, Anna Kręgiel-Rosiak, Krystyna Gawlik

**Affiliations:** Department of Health Sciences, John Paul II University of Applied Sciences in Biala Podlaska, 21-500 Biala Podlaska, Poland

**Keywords:** kinesio taping, rigid taping, gluteus medius muscle, placebo, sEMG

## Abstract

Background: Gluteus medius muscle (GMed) dysfunctions may be associated with pain and functional problems in the lumbar spine and lower limbs. The study sought to assess the effects of applying kinesio taping (KT) and rigid taping (RT) on GMed in the dominant leg of healthy individuals. Furthermore, an attempt was made to indicate which of the applied exercises brought about the greatest activation of GMed. Methods: The study included 90 individuals, with an average age of 21.79. The participants were randomly assigned to one of three groups: kinesio tape (KT), rigid tape (RT) and placebo tape (C). GMed activation was assessed using sEMG during the performance of such exercises as glute bridge, unilateral glute bridge, clamshell, pelvic drop and lunge. Each of the participants was examined three times—before taping, immediately after and 48 h after taping. Results: Before taping, the greatest GMed activation on the dominant side was noted in clamshell (54.12 %MVIC), whereas the lowest GMed activation was observed in glute bridge (36.35 %MVIC). The comparison of results obtained before and immediately after taping in all the groups revealed a statistically significant increase in GMed activation (*p* < 0.05), while the comparison of results achieved before and 48 h after taping showed significant differences in glute bridge in groups KT and RT. In all the groups, the differences in results obtained in the other exercises were not significant. Taking into account each of the applied exercises, at none of the three stages of examination were the differences between the groups significant. Conclusions: Regardless of the type of taping applied (KT, RT, C), a significant increase in GMed activation was noted in all the exercises immediately after taping. At none of the stages of examination were the differences between the groups significant.

## 1. Introduction

Gluteus medius muscle (GMed) is the main hip abductor that ensures stability in the coronal plane of the pelvis while walking and performing other functional activities [1,2]. Gluteus muscle dysfunctions may be associated with pain and functional problems in the lumbar spine as well as in hip, knee and ankle joints [3]. There is evidence that the above-mentioned symptoms are clinically related to gluteus muscle atrophy or weakness [4]. Therefore, it is important to understand the most effective methods for activating gluteus muscles [5]. One of them is taping applied in the form of kinesio taping (KT) and rigid taping (RT).

KT was developed by Kenso Kase in the 1970s. Since then, it has been used in the treatment of various lesions [6,7,8,9]. Kinesio tape is a narrow elastic strip that can stretch up to 120–140% of its original length, thus, helping to reduce movement limitation. According to the method developers, the use of KP in the form of applying the tape from the origin to the insertion of a muscle may strengthen a reaction from the muscle spindle and facilitate muscle contraction [10]. This idea is based on a previous neurological study, which revealed that cutaneous afferent signals, presumably linked with proprioceptors, modify excitability of slow and fast motor units in a variety of ways [11]. It was also stated that the use of KT activates cutaneous mechanoreceptors, thus, alleviating pain (according to the gate control theory assumptions) [10].

Scientific reports on the effects of KT on the human body are contradictory. Although several mechanisms of KT effectiveness have been put forward, the exact mechanism has not been discovered yet. It is believed that KT may change proprioceptive information from peripheral tissues and improve neuromuscular control [12,13]. According to Ye [14], KT may increase muscle strength. Other researchers claim that positive effects of kinesio taping may stem solely from the placebo effect [15,16,17,18,19]. Kinesio taping is considered to prevent sports-related injuries and reduce muscle cramps, swelling and pain [20,21]. However, Reneker et al. [22] pointed to the lack of convincing evidence that KT contributes to enhancing performance in athletes. Moreover, to date, limited evidence has supported the use of KT to increase muscle strength. There is also a scarcity of data on the effectiveness of treating musculoskeletal conditions with kinesio taping [23].

Rigid tape is made of cotton and it has no elasticity [24]. It is mainly used in sports-related ankle injuries, patellofemoral pain and shoulder conditions [21,25,26,27]. RT stabilizes the taped body part, changes pain perception and improves balance performance [28,29]. Similar to KT, research findings on RT effects are equivocal [30,31]. Macgregor et al. [30] noted an increase in vastus medialis obliquus (VMO) activation following RT; however, Alexander et al. [31] did not confirm such an effect for the trapezius muscle.

The present study sought to assess the effects of applying kinesio taping (KT), rigid taping (RT) and placebo taping (C) on gluteus medius muscle activation in the dominant leg of healthy individuals. Furthermore, an attempt was made to indicate which of the applied exercises brought about the greatest activation of GMed.

## 2. Materials and Methods

### 2.1. Study Design

This single-blinded, controlled, randomized, clinical study was conducted in accordance with the Declaration of Helsinki and it was approved by the Bioethics Committee of John Paul II University of Applied Sciences (ABNS) in Biala Podlaska (protocol code 5/2019, 5 June 2019) and registered at Clinical Trials (NCT04345224).

All the participants were informed about the purpose of the study and that they could quit the study at any time. Moreover, they gave their written informed consent to for their data to be collected during the study for scientific purposes (publishing).

### 2.2. Participants

The study included 90 healthy Caucasian students. The inclusion criteria were as follows: the absence of pain or any signs of musculoskeletal dysfunctions in the lower limbs, no history of surgery and/or orthopedic dysfunctions in the lower limbs in the previous year, the absence of cardiovascular or respiratory diseases that impeded the performance of the movements under study, the absence of systemic or vestibular diseases that affected balance and the absence of peripheral, central metabolic or neurological diseases. The exclusion criteria were as follows: the inability to perform any of the assessment and/or intervention procedures, the presence of considerable musculoskeletal pain or discomfort during the performance of any assessment and/or intervention procedures and the presence of allergic reaction to the tape application.

A total of 101 individuals volunteered to participate in the study. However, 11 persons were excluded from the study because they did not meet the inclusion criteria. Five more participants were excluded from the final stage, i.e., examination 48 h after taping, due to allergic reactions or tape loss (Figure 1).

### 2.3. Procedures

We performed a single-blind (subject) study with repeated-measures design to evaluate the effects of KT and RT on gluteus medius muscle activation. The control group (C) consisted of individuals who received placebo taping. In each group, the tape was applied on the dominant side.

The tape application was randomized with opaque sealed envelopes. Each participant drew one envelope from an opaque box, opened the envelope and read out the group symbol (KT, RT, C).

The kinesio tape (Kinesio Tex Classic) was applied to the lateral hip with the participant in a side-lying position. The first part of one I strip was attached to the posterior iliac crest without tension and without crossing the target tissue. The participant flexed the adducted hip actively in order to make it possible to apply the middle part of the tape with tension of approximately 50% (Figure 2). Afterward, with the leg in the original position, the remainder of the tape was applied without any tension, ending approximately at the greater trochanter. Next, the second I strip was applied in the same manner starting at the anterior iliac crest [32]. The tension of the tape (50%) was calculated according to the following formula:(L − 4/1.50) + 4
where L is the length of the tape, 4 represents the length of the anchors (2 cm in each end) and 1.50 depicts the required tension [33].

All the participants were instructed to leave the tape in place until the follow-up visit.

The same procedure was used in the group that received the rigid tape (Endura Sports Tape). The only difference was that there was no tension in the tape (Figure 3). The participants were instructed to leave the tape in place until the follow-up visit.

The control group (placebo tape application) received a single strip of paper tape (Endura Fix Tape) across the lateral affected hip without tension in the tape or muscle stretch (Figure 4) [32]. The participants were instructed to leave the tape in place until the follow-up visit.

Each participant was examined three times—before taping, immediately after and 48 h after taping.

### 2.4. Electromyographic Measurements

Prior to placing electrodes on the body, the skin was cleaned with 90% alcohol solution. Ag/Ag electrodes 30 mm in diameter and with a conducting area of 16 mm (SORIMEX, Torun, Poland) were placed in accordance with the standards of Surface Electromyography for Non-invasive Assessment of Muscles (SENIAM). The electrodes were placed halfway between the iliac crest and the greater trochanter of the femur.

The electrodes were placed along muscle fibers of GMed by an experienced physical therapist (K.Z.—author). All sEMG examinations were performed between 8 a.m. and 12 a.m. using 8-channel Naroxon Ultium EMG system and mioMUSCLE system (Naroxon U.S.A. Inc. Scottsdale, AZ, USA).

The examination began with an EMG assessment in the resting condition. The aim was to find out what the baseline state of GMed activation in a supine position was. Afterwards, MVIC (Maximum Voluntary Isometric Contraction) data were gathered during the manual test of muscle strength, with each participant in a side-lying position and with hip abduction of 20° in the dominant limb. During the MVIC test, three EMG signals were acquired for each participant. The participants performed each maximum GMed contraction for 5 s, with 30 s intervals in between [34].

Exercise order was randomized using a random pattern generator so as to avoid any order bias due to fatigue. The participants were barefoot while performing exercises to prevent any potential variations that may have occurred due to footwear.

Prior to the commencement of the exercises, the participants were instructed how to perform the exercises through verbal instructions and a practical demonstration. During the exercise performance, the examiner did not encourage or support the participants.

Two minutes of rest were given between the performance of each exercise. The participants performed eight repetitions of each exercise, i.e., three practice repetitions and five repetitions that were used for data collection. Exercises were performed to a metronome set at 60 beats per minute to standardize the rate of movement across the participants.

All the data were rectified and smoothed using a root-mean-square algorithm. They were smoothed with a 50 millisecond (msec) time reference. Peak amplitudes were averaged over a 100 msec window of time, i.e., 50 msec prior to the peak and 50 msec after the peak.

The following functional tasks were chosen for the dynamic EMG evaluation: glute bridge, unilateral glute bridge, clamshell, pelvic drop and lunge. A description of each exercise can be found in Appendix A.

The examinations were carried out three times—before taping, immediately after and 48 h after taping (Figure 1).

### 2.5. Data Analyses

Research results were analyzed using IMB SPSS 28.0 (Softonic, Miami Beach, FL, USA). Before carrying out parametric analyses with the use of the T test for dependent samples as well as one-way analysis of variance (ANOVA), adequate assumptions concerning normality of distribution and homogeneity of variance were checked. It should also be stressed that in the case of groups with an equal number of subjects, the aforementioned tests are quite resistant to one of the assumptions (i.e., normality of distribution or homogeneity of variance) when the T test is used. Quantitative data were presented taking into account such descriptive statistics as mean (M) and standard deviation (SD). Comparative analysis between the groups was carried out using the Bonferroni test. Statistical significance was set at *p* ≤ 0.05.

## 3. Results

Mean age of the group under study was 21.79 (±0.94). All the participants were familiarized with the study procedure and provided their written informed consent to take part in it. Women constituted a slightly larger percentage of the study participants (55.6%) (Table 1). Detailed characteristics of the examined groups, taking into consideration the type of taping applied, can be found in Table 2.

Taking into account each of the applied exercises, significant differences between the groups were sought (Table 3).

The analysis of the data presented in Table 3 revealed no statistically significant differences (*p* > 0.05) between the groups in the results (%MVIC) obtained in the applied exercises at particular stages of the examination, i.e., before taping, immediately after and 48 h after taping.

Before taping, the greatest GMed activation on the dominant side was noted in clamshell (54.12 %MVIC), whereas the lowest GMed activation was observed in glute bridge (36.35 %MVIC)—Table 4.

In all the groups under study, the comparison of results obtained before and immediately after taping revealed a statistically significant increase in GMed activation (*p* < 0.05).

The comparison of results achieved before and 48 h after taping showed significant differences in glute bridge in the KT and RT groups. In all the groups, the differences in results obtained in the other exercises were not significant (Table 4).

## 4. Discussion

The study focused on the effects of KT, RT and placebo on GMed activation in healthy individuals. A significant increase in GMed activation was noted in all the groups and in all the exercises immediately after taping. After 48 h, the difference in GMed activation was noted in groups KT and RT in glute bridge only. Importantly, at none of the stages of examination were the differences between the groups significant.

The literature on the subject shows that there is evidence that numerous clinical symptoms of the musculoskeletal system are often related to weakness or atrophy of hip abductors [35]. Improper functioning of GMed is linked with back pain [36], hip instability [37,38] and other pathologies [39,40]. GMed muscles are used in mediolateral stabilization when performing a single-leg stance [41,42]. If the hip is not stable during single-limb activities, the femur may adduct and rotate internally, which changes muscle involvement and kinematics within the knee [43,44]. Therefore, GMed weakness affects movement and activity of vastus medialis obliquus (VMO)/vastus lateralis (VL) in the knee, where both functions of these muscle groups are interconnected. Thus, it is important to define which exercises help to produce the greatest GMed activation, which was also the aim of the present work. The current study showed that before taping, the greatest GMed activation on the dominant side was noted in clamshell, whereas the lowest GMed activation was observed in glute bridge, which is not in line with the findings of other researchers. Moore et al. [45] stated that pelvic drop and single-leg bridge exercises generated the greatest GMed activation. In their literature review, Reiman et al. [46] indicated that, of all the exercises analyzed in our study, pelvic drop was the most effective exercise activating GMed, whereas lunge was the least effective.

How KT and RT work has not been fully explained yet. Macgregor et al. [30] revealed that KT placed above the vastus medialis obliquus muscle stimulates cutaneous afferents, which leads to an increase in muscle activity and speed of motor unit activation. Contrary results were obtained by Alexander et al. [31]. When assessing the effects of taping on the monosynaptic reflex of the trapezius muscle in healthy individuals, these researchers noted a decrease in electromyographic muscle activity by 22%. Another study conducted by the same group of researchers focused on the effects of RT on triceps surae in healthy subjects. Again, they observed a reduction in triceps surae excitability [47].

Kenzo Kase, the developer of the KT method, claimed that the tape may affect muscle activity (motor unit recruitment) through stimulating cutaneous mechanoreceptors. This hypothesis was confirmed by Konishi et al. [48], who noted that tactile stimulation with KT caused an increase in muscle activity associated with Ia afferents. They stated that afferent feedback from mechanoreceptors through sensory activation is sent to gamma motor neurons, which are important for Ia afferent modulation [48]. The findings of our study may confirm this hypothesis; however, according to our results, it does not matter what type of tape is applied and effects are not long lasting. Another possible explanation is the occurrence of the placebo effect. Mak et al. [18] tried to explain this phenomenon. They examined changes in the activation of wrist extensors brought about by KT. These researchers did not find any significant differences between individuals who used KT regularly and those who had never applied it. Interestingly, they noted a significant increase in muscle strength in individuals who used KT regularly, which may point to its placebo effect. However, Miller et al. [32] observed that KT may facilitate GMed activation and improve postural stability in women with unilateral patellofemoral pain syndrome (PFPS).

In the literature, there is a scarcity of data on the effects of taping on GMed activation. Nonetheless, there are studies related to this issue in the context of other muscles. According to de Freitas et al. [49], KT application did not exert any influence on quadriceps muscle activation. Moreover, Halski et al. [50] and Serrão et al. [51] did not note any differences in the activation of vastus medialis (VM), vastus lateralis (VL) and rectus femoris (RF) following KT application. It is also confirmed by Dos Santos Gloria et al. [52]. However, these authors point to the lack of effects of KT on RF activation in healthy female football players.

Cai et al. [53] assessed the effects of facilitatory KT, inhibitory KT and no tape on the activation of wrist extensors. They did not find any differences between the groups under study. Further, Au et al. [54], who examined patients with lateral epicondylitis, did not report any differences in muscle activation between facilitatory KT, inhibitory KT and sham KT groups.

In our study, sEMG was employed to assess KT and RT effectiveness and responses were registered during active muscle contraction. Lins et al. [55] applied KT to RF, VL and VM muscles of the dominant leg in 20 healthy individuals and compared their results with the control group (no tape) and the placebo group. No differences were noted between the groups during concentric and eccentric knee extension. The authors concluded that KT did not change neuromuscular performance of femoral quadriceps.

Other researchers also point to the reduction in muscle activation resulting from KT. When examining VMO and VL activation after KT application, Lee et al. [56] noted its decrease during stair ascent and descent in patients with patellofemoral pain syndrome (PFPS). Ataullach et al. [57] assessed the effects of KT on GMed activation in athletes with chronic ankle instability. They found a significant decrease in GMed activation in group KT and an insignificant decrease in GMed activation in the control group.

In the literature, there are studies that point to an increase in muscle activity caused by KT. Hsu et al. [58] noted increased EMG activity of the lower trapezius muscle in the 60°−30° arm-lowering phase in baseball players with shoulder impingement after KT application compared to the placebo taping. Slupik et al. [59] applied KT to the quadriceps muscle and measured EMG activity during isometric maximal knee extension at different points in time after taping. They observed that KT increased EMG activity of the quadriceps muscle 24 h after taping but the tape application itself did not have a direct influence on muscle strength.

The effects of KT on EMG activity of the triceps surae muscle and on vertical jump performance were measured by Huang et al. [60]. It was noted that KT increased EMG activity of the medial gastrocnemius muscle but did not contribute to an increase in the jumping height.

Briem et al. [61] assessed the effects of KT and RT on the fibularis longus muscle. Their results showed a significant increase in muscle activity only when RT was applied. In our study, we did not note any differences between the tapes applied.

Scientific reports regarding RT are also inconsistent. Sermenli et al. [62] observed that RT did not influence upper trapezius muscle activation, while Cowan et al. [63] noted its impact on VMO and VL activity. There was no such effect in the placebo group.

Due to contradictory reports produced by different researchers, further research is needed in this regard. However, taping can be considered as part of comprehensive rehabilitation plans for pathological conditions and injury prevention [64,65,66].

### Limitations

The current study has some limitations. The examiners were not blinded, which could influence the results. Additionally, the subjects were tested immediately after the taping and 48 h after the intervention. There is a likelihood of obtaining better test results due to the improved physical performance, rather than the intervention used.

In each of the three groups, tapes were applied both on the dominant and non-dominant side. In the study, we only concentrated on results obtained for the dominant side. The comparison of effects that KT and RT had on both sides will be the focus of our next study.

The application of surface electrodes (and not needle electrodes) may have caused some signal disturbances during the gluteus maximus muscle activation.

The study was conducted on healthy individuals. The tapes applied may have a different influence on GMed activation in patients with orthopedic and neurological conditions. Therefore, it is necessary to carry out further research on patients with different health conditions.

## 5. Conclusions

Regardless of the type of taping applied (kinesio tape, rigid tape, placebo), a significant increase in GMed activation was noted in all the exercises immediately after taping. At none of the stages of examination were the differences between the groups significant, which may point to the placebo effect of the tapes.

## Figures and Tables

**Figure 1 ijerph-19-14889-f001:**
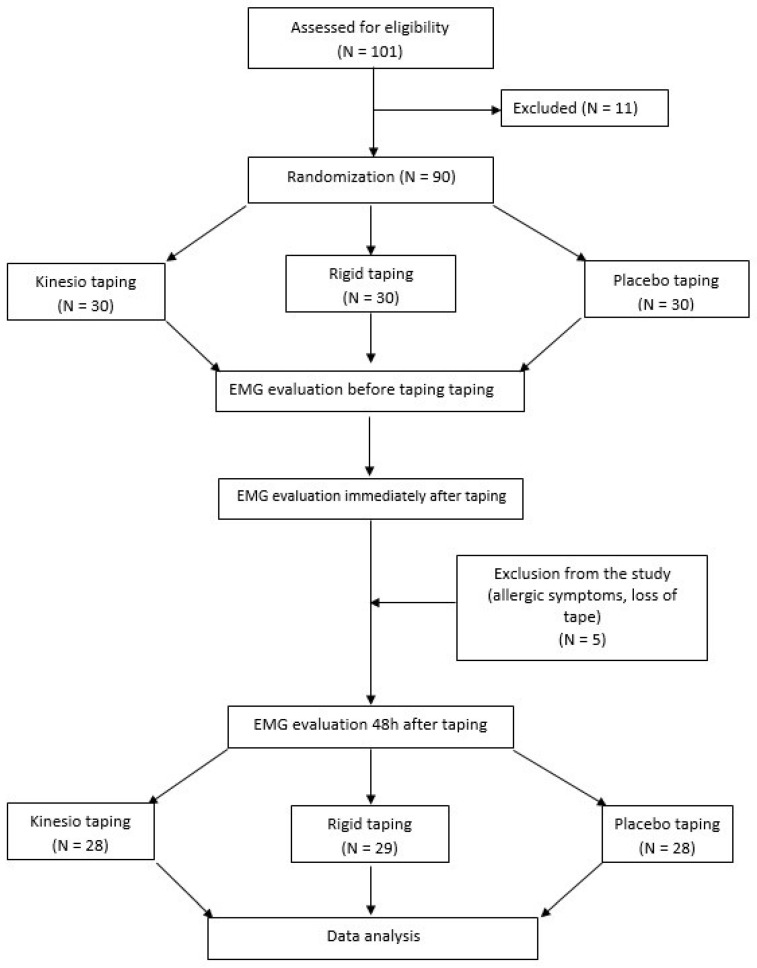
Flow diagram.

**Figure 2 ijerph-19-14889-f002:**
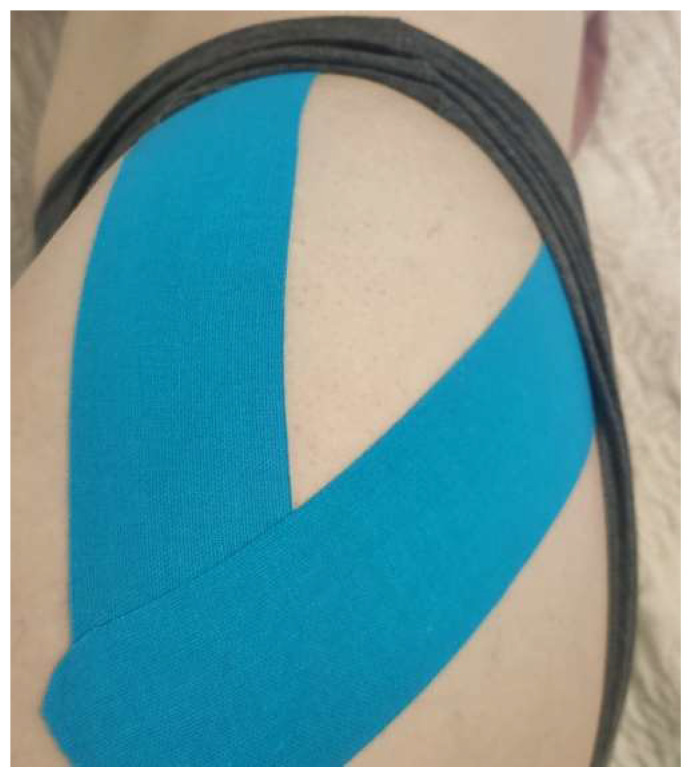
The method of kinesio tape application used.

**Figure 3 ijerph-19-14889-f003:**
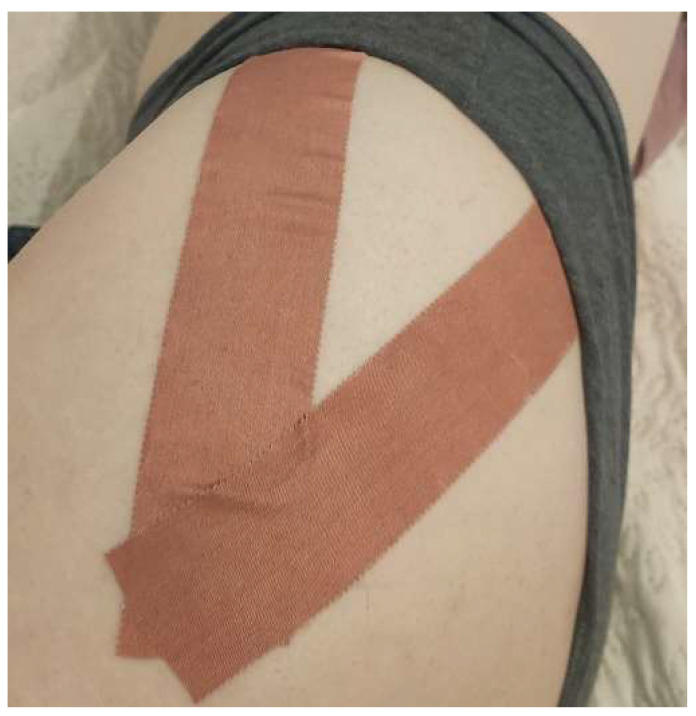
The method of rigid tape application used.

**Figure 4 ijerph-19-14889-f004:**
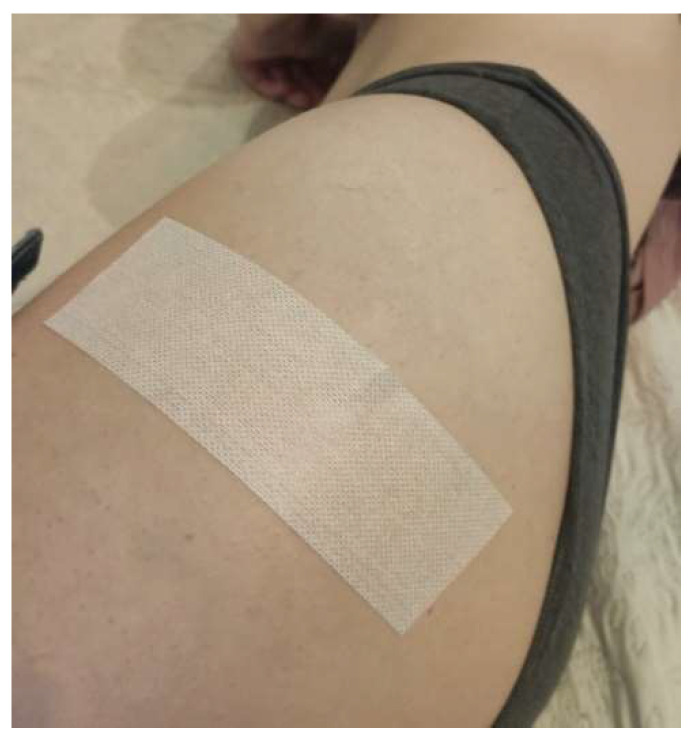
The method of placebo tape application used.

**Table 1 ijerph-19-14889-t001:** Sex distribution in the groups under study (N = 90).

	Group			Total N (%)	V Kramer	Chi2	*p*
	Kinesio Tape N (%)	Rigid Tape N (%)	Placebo N (%)				
Female	18 (60)	18 (60)	14 (46.7)	50 (55.6)	0.126	1.440	0.487
Male	12 (40)	12 (40)	16 (53.3)	40 (44.4)			
Total	30 (100)	30 (100)	30 (100)	90 (100)			

**Table 2 ijerph-19-14889-t002:** Homogeneity of the groups under study (N = 90).

Variable	Group	N	$\bar{x}$ (±SD)	Standard Error	95% Confidence Interval		F	*p*
					Min	Max		
Age	KT	30	21.97 (0.89)	0.16	21.63	22.30	0.950	0.391
	RT	30	21.77 (0.94)	0.17	21.42	22.12		
	C	30	21.63 (1.00)	0.18	21.26	22.01		
Body height	KT	30	169.77 (8.52)	1.55	166.59	172.95	0.313	0.732
	RT	30	168.57 (10.09)	1.84	164.80	172.33		
	C	30	170.47 (9.54)	1.74	166.91	174.03		
Body mass	KT	30	66.83 (11.22)	2.05	62.65	71.02	0.287	0.751
	RT	30	66.53 (12.49)	2.28	61.87	71.20		
	C	30	68.67 (11.69)	2.13	64.30	73.03		
Body Mass Index	KT	30	23.07 (2.46)	0.45	22.16	23.99	0.185	0.832
	RT	30	23.26 (2.79)	0.51	22.22	24.30		
	C	30	23.46 (2.04)	0.37	22.70	24.22		

KT—Kinesio Taping Group, RT—Rigid Taping Group, C—Control Taping Group.

**Table 3 ijerph-19-14889-t003:** Group differences in results obtained for particular exercises (N = 90).

Dependent Variable			Mean Difference (I-J)	Standard Error	Significance	95% Confidence Interval	
						Min	Max
Glute Bridge before taping	KT	RT	−0.08	4.60	1.00	−11.31	11.16
		C	−8.49	4.60	0.21	−19.72	2.74
	RT	KT	0.08	4.60	1.00	−11.16	11.31
		C	−8.41	4.60	0.21	−19.65	2.82
	C	KT	8.49	4.60	0.21	−2.74	19.72
		RT	8.41	4.60	0.21	−2.82	19.65
Glute Bridge immediately after taping	KT	RT	−0.31	4.64	1.00	−11.64	11.02
		C	−9.77	4.64	0.11	−21.10	1.56
	RT	KT	0.31	4.64	1.00	−11.02	11.64
		C	−9.46	4.64	0.13	−20.79	1.87
	C	KT	9.77	4.64	0.11	−1.56	21.10
		RT	9.46	4.64	0.13	−1.87	20.79
Glute Bridge 48 h after taping	KT	RT	−0.04	4.65	1.00	−11.40	11.32
		C	−8.88	4.65	0.18	−20.24	2.48
	RT	KT	0.04	4.65	1.00	−11.32	11.40
		C	−8.84	4.65	0.18	−20.20	2.52
	C	KT	8.88	4.65	0.18	−2.48	20.24
		RT	8.84	4.65	0.18	−2.52	20.20
Unilateral Glute Bridge before taping	KT	RT	0.36	5.23	1.00	−12.42	13.13
		K	−9.16	5.23	0.25	−21.93	3.62
	RT	KT	−0.36	5.23	1.00	−13.13	12.42
		C	−9.51	5.23	0.22	−22.29	3.26
	C	KT	9.16	5.23	0.25	−3.62	21.93
		RT	9.51	5.23	0.22	−3.26	22.29
Unilateral Glute Bridge immediately after taping	KT	RT	−0.05	5.12	1.00	−12.54	12.44
		C	−10.55	5.12	0.13	−23.04	1.94
	RT	KT	0.05	5.12	1.00	−12.44	12.54
		C	−10.50	5.12	0.13	−22.99	1.99
	C	KT	10.55	5.12	0.13	−1.94	23.04
		RT	10.50	5.12	0.13	−1.99	22.99
Unilateral Glute Bridge 48 h after taping	KT	RT	1.93	5.17	1.00	−10.70	14.56
		C	−8.57	5.17	0.30	−21.20	4.06
	RT	KT	−1.93	5.17	1.00	−14.56	10.70
		C	−10.50	5.17	0.14	−23.13	2.13
	C	KT	8.57	5.17	0.30	−4.06	21.20
		RT	10.50	5.17	0.14	−2.13	23.13
Clamshell before taping	KT	RT	3.44	5.31	1.00	−9.52	16.41
		C	−4.42	5.31	1.00	−17.38	8.54
	RT	KT	−3.44	5.31	1.00	−16.41	9.52
		C	−7.86	5.31	0.43	−20.83	5.10
	C	KT	4.42	5.31	1.00	−8.54	17.38
		RT	7.86	5.31	0.43	−5.10	20.83
Clamshell immediately after taping	KT	RT	3.13	5.37	1.00	−9.97	16.23
		C	−7.36	5.37	0.52	−20.46	5.74
	RT	KT	−3.13	5.37	1.00	−16.23	9.97
		C	−10.49	5.37	0.16	−23.59	2.61
	C	KT	7.36	5.37	0.52	−5.74	20.46
		RT	10.49	5.37	0.16	−2.61	23.59
Clamshell 48 h after taping	KT	RT	3.06	5.22	1.00	−9.69	15.81
		C	−6.91	5.22	0.57	−19.66	5.83
	RT	KT	−3.06	5.22	1.00	−15.81	9.69
		C	−9.97	5.22	0.18	−22.72	2.77
	C	KT	6.91	5.22	0.57	−5.83	19.66
		RT	9.97	5.22	0.18	−2.77	22.72
Lunge before taping	KT	RT	−0.34	4.89	1.00	−12.27	11.59
		C	3.81	4.89	1.00	−8.12	15.74
	RT	KT	0.34	4.89	1.00	−11.59	12.27
		C	4.14	4.89	1.00	−7.79	16.07
	C	KT	−3.81	4.89	1.00	−15.74	8.12
		RT	−4.14	4.89	1.00	−16.07	7.79
Lunge immediately after taping	KT	RT	−1.29	4.52	1.00	−12.33	9.75
		C	0.99	4.52	1.00	−10.05	12.03
	RT	KT	1.29	4.52	1.00	−9.75	12.33
		C	2.29	4.52	1.00	−8.75	13.33
	C	KT	−0.99	4.52	1.00	−12.03	10.05
		RT	−2.29	4.52	1.00	−13.33	8.75
Lunge 48 h after taping	KT	RT	0.00	4.33	1.00	−10.57	10.57
		C	5.03	4.33	0.74	−5.53	15.60
	RT	KT	0.00	4.33	1.00	−10.57	10.57
		C	5.03	4.33	0.74	−5.53	15.60
	C	KT	−5.03	4.33	0.74	−15.60	5.53
		RT	−5.03	4.33	0.74	−15.60	5.53
Pelvic Drop before taping	KT	RT	4.24	4.59	1.00	−6.96	15.43
		C	−6.14	4.59	0.55	−17.34	5.05
	RT	KT	−4.24	4.59	1.00	−15.43	6.96
		C	−10.38	4.59	0.08	−21.58	0.82
	C	KT	6.14	4.59	0.55	−5.05	17.34
		RT	10.38	4.59	0.08	−0.82	21.58
Pelvic Drop immediately after taping	KT	RT	4.56	4.64	0.98	−6.76	15.89
		C	−8.01	4.64	0.26	−19.33	3.31
	RT	KT	−4.56	4.64	0.98	−15.89	6.76
		C	−12.57	4.64	0.02	−23.90	−1.25
	C	KT	8.01	4.64	0.26	−3.31	19.33
		RT	12.57	4.64	0.02	1.25	23.90
Pelvic Drop 48 h after taping	KT	RT	5.34	4.57	0.74	−5.80	16.48
		C	−5.48	4.57	0.70	−16.62	5.67
	RT	KT	−5.34	4.57	0.74	−16.48	5.80
		C	−10.82	4.57	0.06	−21.96	0.33
	C	KT	5.48	4.57	0.70	−5.67	16.62
		RT	10.82	4.57	0.06	−0.33	21.96

KT—Kinesio Taping Group, RT—Rigid Taping Group, C—Control Taping Group.

**Table 4 ijerph-19-14889-t004:** Comparison of results (%MVIC) in the exercises performed at particular stages of examination (N = 90).

Test	Group	I—Before Taping				II—Immediately After Taping				I vs. II		III—48 h After Taping				I vs. III	
		$\bar{x}$ (±SD)	Mean Standard Error	Min	Max	$\bar{x}$ (±SD)	Mean Standard Error	Min	Max	t	*p*	$\bar{x}$ (±SD)	Mean Standard Error	Min	Max	t	*p*
Glute Bridge	KT	33.50 (16.19)	2.96	27.45	39.54	36.47 (15.38)	2.81	30.73	42.22	−3.902	0.001 *	35.14 (15.83)	2.89	29.23	41.05	−2.226	0.034 *
	RT	33.57 (18.39)	3.36	26.71	40.44	36.78 (18.34)	3.35	29.94	43.63	−5.098	0.001 *	35.18 (18.39)	3.36	28.32	42.05	−2.153	0.040 *
	C	41.99 (18.79)	3.43	34.97	49.00	46.24 (19.91)	3.63	38.81	53.67	−5.872	0.001 *	44.03 (19.64)	3.59	36.69	51.36	−1.324	0.196
	Σ	36.35 (18.07)	1.91	32.57	40.14	39.83 (18.35)	1.93	35.99	43.67	−2.87	0.005 *	38.12 (18.31)	1.93	34.28	41.95	−2.87	0.005 *
Unilateral Glute Bridge	KT	45.09 (20.49)	3.74	37.43	52.74	48.08 (19.84)	3.62	40.67	55.48	−2.422	0.022 *	47.56 (20.51)	3.74	39.90	55.22	−1.670	0.106
	RT	44.73 (20.42)	3.73	37.11	52.35	48.13 (19.53)	3.56	40.84	55.42	−2.374	0.024 *	45.63 (20.06)	3.66	38.14	53.12	−0.558	0.581
	C	54.24 (19.87)	3.63	46.82	61.66	58.63 (20.10)	3.67	51.12	66.13	−4.728	0.001 *	56.13 (19.53)	3.57	48.84	63.43	−1.878	0.071
	Σ	48.02 (20.52)	2.16	43.72	52.32	51.61 (20.22)	2.13	47.37	55.85	−5.15	0.001 *	49.78 (20.34)	2.14	45.52	54.03	−2.20	0.030 *
Clamshell	KT	53.79 (21.69)	3.96	45.69	61.89	56.11 (22.24)	4.06	47.81	64.42	−2.741	0.010 *	54.90 (22.00)	4.02	46.68	63.11	−1.429	0.164
	RT	50.35 (18.64)	3.40	43.39	57.31	52.98 (19.93)	3.64	45.54	60.42	−2.994	0.006 *	51.84 (19.78)	3.61	44.45	59.22	−1.086	0.287
	C	58.21 (21.23)	3.88	50.28	66.14	63.47 (20.11)	3.67	55.96	70.98	−5.384	0.001 *	61.81 (18.75)	3.42	54.81	68.81	−2.141	0.061
	Σ	54.12 (20.59)	2.17	49.80	58.43	57.52 (21.02)	2.22	53.12	61.93	−6.38	0.001 *	56.18 (20.43)	2.15	51.90	60.46	−2.69	0.009 *
Pelvic Drop	KT	41.99 (17.81)	3.25	35.34	48.64	44.52 (18.35)	3.35	37.66	51.37	−4.044	0.001 *	44.31 (17.69)	3.23	37.70	50.92	−3.485	0.002 *
	RT	37.75 (15.89)	2.90	31.82	43.69	39.95 (16.12)	2.94	33.94	45.97	−3.012	0.005 *	38.97 (15.98)	2.92	33.00	44.94	−1.443	0.160
	C	48.13 (19.42)	3.55	40.88	55.39	52.53 (19.27)	3.52	45.33	59.72	−5.875	0.001 *	49.79 (19.22)	3.51	42.61	56.96	−1.529	0.137
	Σ	42.63 (18.08)	1.91	38.84	46.41	45.67 (18.51)	1.95	41.79	49.54	−7.34	0.001 *	44.36 (18.04)	1.90	40.58	48.13	−3.43	0.001 *
Lunge	KT	44.09 (20.67)	3.77	36.37	51.81	46.39 (19.84)	3.62	38.98	53.80	2.015	0.011 *	46.16 (19.81)	3.62	38.76	53.55	−1.303	0.203
	RT	44.43 (18.73)	3.42	37.44	51.42	47.68 (17.15)	3.13	41.28	54.09	−1.787	0.084	46.16 (16.90)	3.09	39.85	52.47	−1.379	0.179
	C	40.29 (17.23)	3.15	33.85	46.72	45.40 (15.24)	2.78	39.71	51.09	45.40	0.001 *	41.12 (12.86)	2.35	36.32	45.92	−0.510	0.614
	Σ	42.94 (18.81)	1.98	39.00	46.88	46.49 (17.34)	1.83	42.86	50.12	−4.07	0.001 *	44.48 (16.75)	1.77	40.97	47.99	−1.80	0.076

*—significant differences at *p* < 0.05, KT—Kinesio Taping Group, RT—Rigid Taping Group, C—Control Taping Group, Σ—Total.

## Data Availability

Not applicable.

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
