# Peer review of "Effects of Kinesio Taping and Rigid Taping on Gluteus Medius Muscle Activation in Healthy Individuals: A Randomized Controlled Study"

_ijerph, 2022, doi:10.3390/ijerph192214889_

Round 1
Reviewer 1 Report
First of all, congratulations to the authors for the article.
I present the suggestions for the article:
In line 97 where the percentage of women is discussed, I recommend the presentation of total data for both sexes.
Procedures section.
There are several ways of performing gluteus medius facilitation, why don't they use Y-facilitation?
How did you measure the tension of the bandage? It is important to introduce it, there are studies that show that depending on the brand they have different lengths depending on the tension.
When you comment without tension, I understand that it would be the paper off method (this section would be interesting to introduce, since if it is applied directly when the strip is removed, it has a slight tension).
Electromyographic procedure.
It is not explained how the electrodes are placed, since there is a standardized way for their placement, to avoid as much as possible the contraction of the gluteus maximus.
Statistical analysis.
Were data normality tests performed (the hypothesis of normality by Kolmogorov-Smirnov)? In addition, homogeneity of variance between groups was assessed by Levene's test).
The results with Student's t test, for example, are not the most correct:
To analyze the effect, analysis of covariance (ANOVA).
Effect size (ES), Cohen's calculations.
Bonferroni correction to analyze the association between effects.
Plagiarism report:
Observed on line 113 to 118, 139 to 158 and appendix A.
The related articles are:
Boren K, Conrey C, Le Coguic J, Paprocki L, Voight M, Robinson TK. Electromyographic analysis of gluteus medius and gluteus maximus during rehabilitation exercises. Int J Sports Phys Ther. 2011 Sep;6(3):206-23
Silva RO, Carlos FR, Morales MC, Emerick VS, Teruyu AI, Valadão VMA, Carvalho LC, Lobato DFM. Effect of two Dynamic Tape™ applications on the electromyographic activity of the gluteus medius and functional performance in women: A randomized, controlled, clinical trial. J Bodyw Mov Ther. 2021 Jan;25:212-217. doi: 10.1016/j.jbmt.2020.11.018.
Author Response
Thank you very much for all your comments. All the proposed changes increase the value of our manuscript.
Point 1: In line 97 where the percentage of women is discussed, I recommend the presentation of total data for both sexes.
Response 1: Changes have been introduced into the text and marked in red.
Point 2: There are several ways of performing gluteus medius facilitation, why don't they use Y-facilitation?
Response 2: The manner of applying the tape that was used in the study made it possible to place the electrodes in accordance with the recommendations of SENIAM (www.seniam.org), at the same time covering the bigger area of the facilitated muscle. The description of the electrode location has been added to the text and marked in red.
Point 3: How did you measure the tension of the bandage? It is important to introduce it, there are studies that show that depending on the brand they have different lengths depending on the tension.
Response 3: The tension of the tape (50%) was calculated according to the following formula:
(L – 4 / 1.50) + 4
where L is the tape length, 4 represents the length of the anchors (2 cm in each end), and 1.50 depicts the required tension.
Point 4: When you comment without tension, I understand that it would be the paper off method (this section would be interesting to introduce, since if it is applied directly when the strip is removed, it has a slight tension).
Response 4: Both in the case of the rigid tape (Endura Sport Tape) and placebo (Endura Fix Tape), the tapes that were used cannot be stretched.
Point 5: It is not explained how the electrodes are placed, since there is a standardized way for their placement, to avoid as much as possible the contraction of the gluteus maximus.
Response 5: In accordance with the response to comment 2, we followed the recommendations of SENIAM (www.seniam.org) when placing the electrodes. The description of their location has been added to the text and marked in red.
Point 6: Were data normality tests performed (the hypothesis of normality by Kolmogorov-Smirnov)? In addition, homogeneity of variance between groups was assessed by Levene's test).
Response 6: The tests of normality of distribution were carried out. The statistical methods section has been completed.
Point 7: The results with Student's t test, for example, are not the most correct:
To analyze the effect, analysis of covariance (ANOVA).
Effect size (ES), Cohen's calculations.
Bonferroni correction to analyze the association between effects.
Response 7: Thank you very much for pointing it out to us. Changes have been made to the text and marked in red.
Point 8: Plagiarism report:
Observed on line 113 to 118, 139 to 158 and appendix A.
Response 8: Changes have been made and marked in red.

Reviewer 2 Report
Thank you for letting me review this interesting manuscript about the effects of KT, RT and placebo tape in the gluteus medius activation.
After a careful evaluation, I agree in most part with the conclusion of the authors and I appreciate the effort that all of them have made. However, in my opinion, this manuscript needs extensive revision and some question should be adressed. Otherwise, I would not recommend the publication.
- Line 54. abbreviations should be checked through the entire manuscript.
-Line 60-64.The explanation of the rigid tape should be concordant with the explanation of the KT.
- Line 65. The aim shoud be clear. If the authors compared 3 techniques, all the techniques should be described in the aim of the study.
- Line 95 to 98 and table 1 and 2. The information provided in these parts should be included in results insted of material and methods. Please revise the guide for authors of the Journal.
- Line 105-107. This information should be included in the paragraph about the ethical aspects.
- Line 110. Why did not you blind the examners/assessors? This is a high risk of bias in this type of studies.
- Line 113. The concealled allocation should be described in detail next to the random allocation information.
- Line 118. How was the tension measured?
- Line 128. Only the participants in the control group were instructed to leave the tape until the follow-up? or all the groups?
- Line 130. A figure explaining the different types of taping is needed.
- Line 140. MVIC. The abbreviation is not explained in the manuscript.
- Line 162. To calculate qualitative data? I guess you mean to calculate differences between groups in qualitative variables?
- Line 166. The normal or non-normal distribution of the variables was not consider? Why did not assess differences between the three groups with a repeated measures ANOVA?
- Line 169. In general the results are not well-presented. Taking into consideration the main objective of the study, the relevant information is the between-groups comparison. So. It should be explained clearly in detail that no differences were found. In this way, the within-groups comparisons are not useful.
- Several risk of bias could influence the inmmediate or 48h of follow-up within-changes. Firstly, the examiners were not blinded which could influence the results. Secondly, there is no explanation about how was instructed the patients to perform the test, if the xaminers encourage them or not. Third, taking into consideration that the measurements were performed immediately after the tape intervention, and 48h of follow-up, the patients could improve the test performing. So, the changes could be related to a better performance insted of the real intervention.
- Line 290. This is not a limitation of the study. This could be considered a salami publication.
- Line 297. The limitations that I pointed out in previous comments must be included.
- Appendix A should include figures with the text.
-
Author Response
Thank you very much for all your comments. All the proposed changes increase the value of our manuscript.
Point 1: Line 54. abbreviations should be checked through the entire manuscript.
Response 1: All the abbreviations have been checked through the entire manuscript.
Point 2: Line 60-64. The explanation of the rigid tape should be concordant with the explanation of the KT.
Response 2: The description has been extended. Changes to the text have been marked in red.
Point 3: Line 65. The aim should be clear. If the authors compared 3 techniques, all the techniques should be described in the aim of the study.
Response 3: The description has been extended. Changes to the text have been marked in red.
Point 4: Line 95 to 98 and table 1 and 2. The information provided in these parts should be included in results instead of material and methods. Please revise the guide for authors of the Journal.
Response 4: The suggested changes have been made and marked in red.
Point 5: Line 105-107. This information should be included in the paragraph about the ethical aspects.
Response 5: The information has been moved.
Point 6: Line 110. Why did not you blind the examiners/assessors? This is a high risk of bias in this type of studies.
Response 6: Blinding was only possible at the first stage of the examination, i.e. before applying the tapes. At further stages, the examiner could see what type of the tape was applied when placing sEMG electrodes.
Point 7: Line 113. The concealled allocation should be described in detail next to the random allocation information.
Response 7: The description has been extended. Changes to the text have been marked in red.
Point 8: Line 118. How was the tension measured?
Response 8: The tension of the tape (50%) was calculated according to the following formula:
(L – 4 / 1.50) + 4
where L is the tape length, 4 represents the length of the anchors (2 cm in each end), and 1.50 depicts the required tension.
Point 9: Line 128. Only the participants in the control group were instructed to leave the tape until the follow-up? or all the groups?
Response 9: Participants from all the groups were asked to leave the tape in place until the follow-up. Thank you for pointing it out to us. The description has been extended.
Point 10: Line 130. A figure explaining the different types of taping is needed.
Response 10: It has been added.
Point 11: Line 140. MVIC. The abbreviation is not explained in the manuscript.
Response 11: The changes suggested have been made and marked in red.
Point 12: Line 162. To calculate qualitative data? I guess you mean to calculate differences between groups in qualitative variables?
Response 12: The description of statistical methods has been corrected. Changes have been marked in red.
Point 13: Line 166. The normal or non-normal distribution of the variables was not consider? Why did not assess differences between the three groups with a repeated measures ANOVA?
Response 13: There were mistakes in the description of statistical methods. Thank you very much for pointing it out to us. Changes to the text have been marked in red.
Point 14: Line 169. In general the results are not well-presented. Taking into consideration the main objective of the study, the relevant information is the between-groups comparison. So. It should be explained clearly in detail that no differences were found. In this way, the within-groups comparisons are not useful.
Response 14: The description has been extended. Changes to the text have been marked in red.
Point 15 Several risk of bias could influence the inmmediate or 48h of follow-up within-changes. Firstly, the examiners were not blinded which could influence the results. Secondly, there is no explanation about how was instructed the patients to perform the test, if the xaminers encourage them or not. Third, taking into consideration that the measurements were performed immediately after the tape intervention, and 48h of follow-up, the patients could improve the test performing. So, the changes could be related to a better performance insted of the real intervention.
Response 15: Thank you very much for pointing out significant limitations of the study. The remarks have been taken into account when correcting the text.
Point 16: Line 290. This is not a limitation of the study. This could be considered a salami publication.
Response 16: The indicated passage has been removed. Changes to the text have been marked in red.
Point 17: Line 297. The limitations that I pointed out in previous comments must be included.
Response 17: Changes to the text have been marked in red.
Point 18: Appendix A should include figures with the text.
Response 18: It has been added.

Round 2
Reviewer 1 Report
The authors are to be congratulated for their efforts after the revisions made.
After reading, I recommend that the KINESIOTAPE tension be reproducible per cm measurement (i.e. Base to end branch) rather than the formula presented, so that there is no bias between patients (as each patient will have different anatomical formations)..
Author Response
Thank you again very much for all your comments. All the proposed changes increase the value of our manuscript.
Point 1: After reading, I recommend that the KINESIOTAPE tension be reproducible per cm measurement (i.e. Base to end branch) rather than the formula presented, so that there is no bias between patients (as each patient will have different anatomical formations).
Response 1: Thank you for your comment. However, we decided to leave the formula pattern that we used. It allows for individual selection of the length of the tape depending on the anthropometric sizes of the person. This way of selecting the length of the tape allowed the objective use of always the same tape tension (50%).

Reviewer 2 Report
I would like to congratulate the authors for the effort taking into consideration all my appreciations. However, there are still some issues to adress:
- Data analysis is still unclear. Was a repeated measures ANOVA used for multiple comparisons?
- In the results paragraph. THe between-groups comparisons should be the first results to present according to the aim of the study. Please clarify that.
- The limitations I exposed in my previous Review report have not been taken into consideration : "Firstly, the examiners were not blinded which could influence the results. Secondly, there is no explanation about how was instructed the patients to perform the test, if the xaminers encourage them or not. Third, taking into consideration that the measurements were performed immediately after the tape intervention, and 48h of follow-up, the patients could improve the test performing. So, the changes could be related to a better performance insted of the real intervention."
Author Response
Once again, thank you very much for all your comments. All the proposed changes increase the value of our manuscript.
Point 1: Data analysis is still unclear. Was a repeated measures ANOVA used for multiple comparisons?
Response 1: The description of statistical methods has been corrected. Changes have been marked in red.
“Was a repeated measures ANOVA used for multiple comparisons?”
No, the T-test for dependent samples was used because the premise of the analyses was to independently see which results in a time comparison yielded the clearest results, on the assumption that we would see greater improvements in performance over time. Thus, the T-test for dependent samples was deliberately and consciously used, omitting pairwise (post-hoc) comparisons because it was less important to compare all pairs. Of course, it was possible to do contrasts as well, but in the end we opted for the T-test for dependent samples, and although the correction for the number of comparisons was not applied here (there were few of them) it is not aggravating due to the fact that, as written earlier, it was important to indicate whether the best results would be obtained after the longest period.
Point 2: In the results paragraph. The between-groups comparisons should be the first results to present according to the aim of the study. Please clarify that.
Response 2: The suggested changes have been made and marked in red.
Point 3: The limitations I exposed in my previous Review report have not been taken into consideration : "Firstly, the examiners were not blinded which could influence the results. Secondly, there is no explanation about how was instructed the patients to perform the test, if the examiners encourage them or not. Third, taking into consideration that the measurements were performed immediately after the tape intervention, and 48h of follow-up, the patients could improve the test performing. So, the changes could be related to a better performance insted of the real intervention."
Response 3: Thank you very much for pointing out significant limitations of the study. The suggested changes in “The Limitations” have been made and marked in red.
Paragraph "2.4. Electromyographic measurements" contains the information “Prior to the commencement of the exercises, the participants were instructed how to perform the exercises through verbal instructions and a practical demonstration.”
In paragraph "2.4. Electromyographic measurements", we added the information "During the exercise performance, the examiner did not encourage or support the participants".
